# Synthesis, Biological Evaluation and Docking Studies of Ring-Opened Analogues of Ipomoeassin F

**DOI:** 10.3390/molecules27144419

**Published:** 2022-07-10

**Authors:** Sarah O’Keefe, Pratiti Bhadra, Kwabena B. Duah, Guanghui Zong, Levise Tenay, Lauren Andrews, Hayden Schneider, Ashley Anderson, Zhijian Hu, Hazim S. Aljewari, Belinda S. Hall, Rachel E. Simmonds, Volkhard Helms, Stephen High, Wei Q. Shi

**Affiliations:** 1School of Biological Sciences, Faculty of Biology, Medicine and Health, University of Manchester, Manchester M13 9PT, UK; 2Center for Bioinformatics, Saarland University, 66123 Saarbrucken, Germany; pratiti.bhadra@bioinformatik.uni-saarland.de (P.B.); levise.tenay@gmail.com (L.T.); aanderson8@bsu.edu (A.A.); 3Department of Chemistry, Ball State University, Muncie, IN 47306, USA; kbduah@iu.edu (K.B.D.); landrews2@bsu.edu (L.A.); hoschneider@bsu.edu (H.S.); 4Department of Chemistry and Biochemistry, University of Maryland, College Park, MD 20742, USA; gzong@umd.edu; 5Department of Biomedical Engineering, Biruni University, 34010 Istanbul, Turkey; 6Feinstein Institute for Medical Research, Northwell Health, 350 Community Dr., Manhasset, NY 11030, USA; zhu1@northwell.edu; 7Ralph E. Martin Department of Chemical Engineering, University of Arkansas, Fayetteville, AR 72701, USA; hsaljewa@uark.edu; 8Department of Microbial Sciences, School of Biosciences and Medicine, University of Surrey, Guildford GU2 7XH, UK; b.s.hall@surrey.ac.uk (B.S.H.); rachel.simmonds@surrey.ac.uk (R.E.S.)

**Keywords:** resin glycosides, macrocyclic natural glycolipids, ring-opened analogues, cytotoxicity, protein translocation, Sec61 translocon, molecular docking

## Abstract

The plant-derived macrocyclic resin glycoside ipomoeassin F (Ipom-F) binds to Sec61α and significantly disrupts multiple aspects of Sec61-mediated protein biogenesis at the endoplasmic reticulum, ultimately leading to cell death. However, extensive assessment of Ipom-F as a molecular tool and a therapeutic lead is hampered by its limited production scale, largely caused by intramolecular assembly of the macrocyclic ring. Here, using in vitro and/or in cellula biological assays to explore the first series of ring-opened analogues for the ipomoeassins, and indeed all resin glycosides, we provide clear evidence that macrocyclic integrity is not required for the cytotoxic inhibition of Sec61-dependent protein translocation by Ipom-F. Furthermore, our modeling suggests that open-chain analogues of Ipom-F can interact with multiple sites on the Sec61α subunit, most likely located at a previously identified binding site for mycolactone and/or the so-called lateral gate. Subsequent in silico-aided design led to the discovery of the stereochemically simplified analogue **3** as a potent, alternative lead compound that could be synthesized much more efficiently than Ipom-F and will accelerate future ipomoeassin research in chemical biology and drug discovery. Our work may also inspire further exploration of ring-opened analogues of other resin glycosides.

## 1. Introduction

Resin glycosides are a large family of plant-derived natural products unique to the morning glory family, Convolvulaceae [1,2]. The vast majority of them contain a macrocyclic ester ring with embedded carbohydrates and are considered active ingredients for many herbal medicines. Despite their distinctive structures and medicinal benefits, the pharmacological properties of most resin glycosides are underexplored. In 2005 and 2007, the ipomoeassin family of resin glycosides was isolated from the leaves of *Ipomoea squamosa* in the Suriname rainforest and exhibited potent cytotoxicity [3,4]. Following a great amount of effort on total synthesis [5,6,7,8] and medicinal chemistry [9,10,11,12], chemical proteomics studies discovered strong inhibition of Sec61-mediated protein translocation as the primary molecular mechanism for ipomoeassin F (Ipom-F, Figure 1) [13], the most potent member of the family. To date, ring expansion [14] and modifications at the *para* position of the cinnamate benzene ring [13] have afforded several analogues with biological activities comparable to or even better than Ipom-F.

The translocation of nascent polypeptides into and across the membrane of the endoplasmic reticulum (ER) is the first and decisive step during the biogenesis of many integral membrane and secretory proteins [15,16,17]. The Sec61 translocon [18,19] is the predominant protein conducting channel at the ER membrane, acting as a dynamic hub to coordinate the translocation of ~one third of the cellular proteome in eukaryotes [17]. While the fidelity of Sec61-mediated protein translocation is essential for proper cellular and organismal function [20,21], small molecule inhibitors that modulate this process have provided valuable insights into the mechanistic complexities of protein translocation at the ER [22,23] and also have potential therapeutic applications [24,25]. In the latter case, small molecule-mediated inhibition of Sec61-dependent protein production shows promise for the clinical treatment of solid tumors (ClinicalTrials.gov: NCT05047536) and presents an attractive, yet underexplored, strategy to mitigate the toxicity associated with the overexpression of *SEC* genes that have been linked to pathogenicity in kidney and liver diseases, diabetes and certain cancers [26].

Small molecule inhibitors typically bind to the central, Sec61α, subunit of the Sec61 translocon [19,22] and the subsequent blockade in Sec61-mediated protein translocation results in potent cytotoxicity that, ultimately, leads to cell death [13,22,27,28]. Besides Ipom-F, the current repertoire of small molecule Sec61 inhibitors includes several other structurally distinct classes of natural products: apratoxins [27], coibamide A [28], cotransins [29], decatransin [30], mycolactone [31,32] and derivatives thereof [24,25]. Although a limited number of synthetic Sec61 inhibitors, e.g., the eeyarestatins [33] and FMP-40139-3 [34], have been identified by library-based screening approaches, the natural products and their derivatives are substantially more potent.

Despite their structural diversity, a common feature shared amongst each of the natural product Sec61 inhibitors is a core, albeit differently sized, cyclic scaffold [24,25]. The integrity of most macrocyclic frameworks appears to confer an essential role for efficient Sec61 inhibition, particularly since two linear analogues of coibamide A showed a significant loss in activity when compared to the cyclic parent compound [35]. Hence, we were surprised when two ring-opened analogues (**1** and **2**, Figure 1) of Ipom-F were discovered to still be active in cytotoxicity assays [9]. In our initial characterization of Sec61α as the cellular target of Ipom-F [13], we additionally used a well-characterized cell-free assay to demonstrate that open-chain analogue **2** efficiently inhibits Sec61-mediated protein translocation in vitro [13]. These preliminary results raised an intriguing hypothesis that the macrocyclic ring may not be essential to the biological activities of Ipom-F, or perhaps resin glycosides in general.

In this report, we present unambiguous evidence using our established in vitro ER membrane translocation assay in combination with live-cell cytotoxicity screening to expand the current scope of ring-modified Ipom-F analogues [9,13,14] and prove that open-chain structures can act as authentic surrogates for Ipom-F. These conclusions are further supported by molecular docking of Ipom-F and analogues within the channel pore of Sec61α [22]. Our modeling also raises the intriguing possibility that several Ipom-F analogues can access multiple binding sites on the Sec61α subunit that likely reflect the stepwise interaction of signal peptides with the Sec61 translocon [36]. We further exploited our in silico modeling studies to design a new and stereochemically simpler open-chain Ipom-F analogue **3** (Figure 1) that we predicted to be an active Sec61 inhibitor. After its synthesis, **3** was experimentally validated to be almost as effective at inhibiting Sec61-mediated protein translocation as Ipom-F and the parent open-chain analogue **2**. Therefore, we present **3** as a new lead compound for the efficient generation of new small molecule inhibitors that can be used to explore the mechanisms of Sec61-mediated protein translocation and potential therapeutic applications.

## 2. Results and Discussion

### 2.1. Synthesis and Cytotoxicity of Nine New Open Chain-Analogues

To investigate the impact of ring-opening, we first submitted the open-chain analogue **1 [9]** to the National Cancer Institute (NCI) 60-cell line screening (Appendix A) and determined the cytotoxicity of nine newly synthesized open-chain analogues **4**–**12** (Figure 2, Appendix A) against MDA-MB-231 cells (Appendix A). When compared to Ipom-F, only a 4–5-fold (NCI 60-cell line: average GI_50_ ∼135 nM versus ∼30 nM) and 9-fold (MDA-MB-231 cells: IC_50_ ~60 nM versus ~7 nM) loss in potency was observed for the ring-opened analogue **1** and, while neither compound induced much cytotoxicity in most ovarian cell lines, both compounds showed a comparable cytotoxic potency towards the majority of breast and melanoma cell lines tested (Appendix A). In MDA-MB-231 cells, and when compared to **1** (IC_50_ ~59 nM), the methyl amide-modified analogue **5** (IC_50_ ~43 nM) showed slightly enhanced activity, analogues **4** (IC_50_ ~168 nM) and **6** (IC_50_ ~205 nM) showed only a marginal loss in activity (3–4 fold) and each of the other six open-chain analogues **7**–**12** showed substantially reduced cytotoxicity (µM range; >35-fold activity loss when compared to **1**). Furthermore, the majority of the nine new open-chain analogues were modestly less cytotoxic towards MDA-MB-231 cells when compared to their direct ring-closed counterparts (analogues **13**–**21** matching with **4**–**12**; Appendix A and Appendix A) [9,11,12,13,14]. Taken together, these data demonstrate that the macrocylic integrity of Ipom-F is not crucial for cytotoxic activity against the majority of cell lines tested and suggest that a similar mechanism likely underlies the cytotoxic activity of open- and closed-chain analogues.

### 2.2. Open-Chain Analogues Selectively Inhibit Sec61-Mediated Protein Translocation In Vitro

Having previously established that the principal molecular basis for the cytotoxicity of Ipom-F is its strong, yet reversible, binding to Sec61α and the resultant wide-ranging blockade of Sec61-mediated protein translocation at the ER [13], we sought evidence that this mechanism also underlies the cytotoxic effects of **1** and other open-chain analogues. Hence, based on our previous characterization of Ipom-F and the open-chain analogue **2** [13], we first used a well-established cell-free translation system supplemented with ER microsomes derived from canine pancreas [37] to study the effects of the open-chain analogues that were the most cytotoxic towards MDA-MB-231 cells (analogues **5** > **1** > **6**; Appendix A) on Sec61-mediated protein translocation in vitro. For comparative purposes, we analyzed the effects of **2** [13] and two closed-chain compounds: Ipom-F and the diester-modified analogue **22** [11] in parallel (Figure 3A).

Following the resolution of radiolabeled proteins by SDS-PAGE (Figure 3B), we used the efficiency with which the N-terminal domain of the type II integral membrane protein Ii (short form of HLA class II histocompatibility antigen gamma chain, isoform 1) was N-glycosylated inside the ER lumen as a robust reporter for the authentic membrane integration of this model Sec61-dependent protein client (Figure 3B, right upper panel, 0Gly versus 2 Gly forms) [13,14,33]. Based on the reduced levels of the N-glycosylated forms of Ii synthesized in the presence of 1 µM of each compound in comparison to the DMSO control (Figure 3B, right upper panel, lanes 3–8 versus lane 1), Ipom-F and analogues **1**, **2**, **5** and **22** efficiently, albeit variably, inhibited the in vitro membrane integration of Ii, while analogue **6** did not (Figure 3B,C).

To further analyze the apparent variations in the potency of ring-modified analogues (Figure 3B,C), we next synthesized Ii in the presence of decreasing (500 µM–5 nM) concentrations of each compound that efficiently inhibited the membrane integration of Ii at 1 µM (Figure 3D). Analyses of these in vitro titrations yielded estimated IC_50_ values in the mid-nanomolar range (Figure 1E), allowing us to rank order compound activity: Ipom-F (IC_50_: 155 nM) > **2** (IC_50_: 202 nM) > **5** (IC_50_: 291 nM) > **22** (IC_50_: 334 nM) > **1** (IC_50_: 562 nM) > **6** (negligible inhibition at 1 µM). Given that the two closed-chain compounds, Ipom-F and **22** (Figure 3A), are, respectively, the most and second least potent active inhibitors of Sec61-mediated protein translocation in vitro, these data suggest that a combination of structural and chemical features, and not merely macrocyclic integrity, are important contributors to the potency of Sec61 inhibition.

### 2.3. Open-Chain Analogues Induce Cytotoxicity Via the Selective Inhibition of Sec61-Mediated Protein Translocation

To confirm that Sec61α is the primary target of open-chain analogues and that this interaction underlies their cytotoxic effects, we next used a resazurin-based cell viability assay (Figure 3F) to compare the effects of each compound on the growth of HCT-116 cells that were wild-type for Sec61α (HCT116 Sec61α-WT) or carrying a heterozygous point mutant in SEC61A1 (HCT116 Sec61α-G80W) that confers resistance to Ipom-F, and reduces binding of mycolactone to the Sec61 translocon by mechanism that involves an alteration in translocon dynamics [22].

Following 72 h treatment with 50 nM of each compound, Ipom-F and analogues **1**, **2**, **5** and **22** induced ~48–64% cell death in HCT116 Sec61α-WT cells, while, as observed in vitro, analogue **6** was the least potent compound tested (~20% cell death; Figure 3G, left). Strikingly, none of the compounds affected the viability of HCT116 Sec61α-G80W cells treated using the same concentration (Figure 3G, right), consistent with Sec61α being their primary molecular target.

To further explore the observed variations in cytotoxicity, we performed cell viability assays using HCT116 Sec61α-WT cells and decreasing (25 µM–1 nM) concentrations of each compound that efficiently caused cell death at 50 nM. Analyses of these in cellula titrations yielded estimated IC_50_ values in the low-to-mid-nanomolar range (Figure 3H) allowing us to, once again, rank order compound activity. Although, as typical for Sec61 inhibitors, the IC_50_ values derived via cytotoxicity are consistently lower than those obtained by analyzing membrane insertion (see Appendix A) [13,14], the cytotoxic potency of each compound closely mirrored the rank order of activity that was observed in vitro. Taken together, these data strongly suggest that, similar to the closed-chain Ipom-F [13] and the diester analogue **22**, the interaction of active open-chain analogues (**1**, **2** and **5**) with Sec61α and the resultant inhibition of Sec61-mediated protein translocation at the ER underlies their cytotoxic effects.

### 2.4. Molecular Docking of Open-Chain Analogues within the Channel Pore of Sec61α Reveals Multiple Binding Sites

We have previously postulated that Ipom-F most likely occludes membrane access via the Sec61 lateral gate [23,38], as recently established for mycolactone [22]. Since the G80W mutation that is located in the transmembrane (TM) helix TM2 of Sec61α confers resistance to mycolactone [22], Ipom-F [22] and each of Ipom-F analogues tested here (Figure 3), while the Ipom-F-Sec61α binding site is yet to be elucidated, we used the cryogenic-electron microscopy (cryo-EM)-derived structure of the canine Sec61 channel bound to mycolactone as a model to explore the potential interaction sites and putative docking conformations of active closed-chain (Ipom-F and **22**) and open-chain analogues (**1**, **2** and **5**) in the previously defined inhibited state of the Sec61α channel pore [22].

These docking studies suggest that Ipom-F occupies the same groove between the TM helices TM2, TM7, TM8 and the cytosolic loop (CL) 4 of Sec61α (Figure 4B) that mycolactone was found to bind in [22], and that was approximately recovered in a previous docking analysis of mycolactone [39] using a similar docking protocol as used here. The predicted binding affinity of Ipom-F is −7.82 ± 0.2 kcal/mol and in this orientation Ipom-F may preferentially interact with the C-terminus of TM2 (Leu89-Ala97), the Gln170-Gly172 region of CL4 (loop between TM4-TM5) and the Trp379-Val382 region of CL8 (loop between TM8-TM9) (Appendix A). The *D*-fucose region of Ipom-F most likely occupies the volume between the TM2 helix and CL4 (Figure 4B), since this region formed a hydrogen bond with side-chain or backbone atoms of Gln170 in CL4 in all of our independent final docking simulation results. Appendix A illustrates the polar and non-polar contacts between the docked Ipom-F derivatives and Sec61α residues in their energetically most favorable binding poses.

Strikingly, and in contrast to Ipom-F, the closed-chain analogue **22** was predicted to bind in two different locations (Figure 4A): either in the same groove where Ipom-F and mycolactone bind or in the upper part of the lateral gate (Figure 4C), with predicted binding affinities of −8.81 ± 0.63 and −7.35 ± 0.23 kcal/mol, respectively. Similar to Ipom-F, analogue **22** interacts with the C-terminus of the TM2 helix, the CL4 region and CL8 (Appendix A). However, when positioned in the upper part of the lateral gate, analogue **22** preferentially interacts with TM7 (Gln294-Val298) (Appendix A).

Similar to the closed-chain analogue **22**, the open-chain analogue **2** was also predicted to bind in two different locations within the Sec61 translocon; namely, in the binding groove of mycolactone [22] and the middle of the lateral gate with very similar predicted binding affinities of −4.5 ± 0.75 and −4.4 ± 0.55 kcal/mol, respectively. As observed for Ipom-F and its closed-chain analogue **22**, the open-chain analogue **2** also strongly interacts with TM2, CL4 and CL8 when occupying the mycolactone binding site identified by cryo-EM (Appendix A). Similar to Ipom-F, analogue **2** is also inclined to hydrogen bond with the CL4 region (Lys171), and this hydrogen bond was identified in four out of five independent docking simulations. However, when occupying the middle of the lateral gate, analogue **2** likely interacts with the plug region (Ile68), TM3 and TM7 (Appendix A).

In contrast, analogue **5** was predicted to bind only at the groove of the Ipom-F binding site with a predicted binding affinity of −3.28 ± 0.9 kcal/mol, where it interacts with the C-terminus of TM2, CL4 and CL8 (see Appendix A). Likewise, the open-chain analogue **1** only binds in one position; at the lower part of the lateral gate (Figure 4F), with a predicted binding affinity of −4.25 ± 0.6 kcal/mol, where it strongly interacts with the plug region (Met65-Ile68), TM3 (Ile123-Gln127) and TM7 (Gln294-Val298) (see Appendix A).

These docking studies offer several new insights into small molecule-mediated inhibition of Sec61. Firstly, they suggest that Ipom-F and the majority of the modeled Ipom-F analogues (**2**, **5** and **22**) likely bind within the same groove as mycolactone (between TM2, TM7, TM8 and CL4 of Sec61α) [22]. Secondly, since certain compounds also show favorable binding affinities at the middle (**2**) or upper part of the lateral gate (**22**) or an exclusive putative docking conformation at the lower part of the lateral gate (**1**), our studies reveal multiple putative binding sites for Ipom-F analogues. Since molecular docking of the (about 10–20 amino acids long) hydrophobic core-portions of signal peptides suggests that these regions prefer to bind in the vicinity of the lateral gate of Sec61α of the Sec61 complex [40], we speculate that Ipom-F analogues may perturb signal peptide binding at more than one site on the Sec61 translocon, and that the flexibility of open-chain analogues may even enhance their ability to perturb the interactions of signal peptides at multiple interaction sites within the Sec61α subunit, thereby influencing their potency and/or substrate selectivity. Finally, since the number of contact residues within the CL4 region reflected the rank order of compound activity observed in vitro and in cellula: Ipom-F = **2** ≥ **5** > **22** > **1** (CL4 contact residues: ~77%, 77%, 62%, 31%, 0% respectively; Appendix A), we propose that the interaction of compounds with the CL4 region is important for the potency of Sec61 inhibition. We further propose that the potential of Ipom-F and **2** to form hydrogen bonds with residues Gln170 and Lys171 in the CL4 region (Appendix A) may provide the molecular basis for their increased potency when compared to other analogues.

### 2.5. Biology-Directed and In Silico-Aided Design of Analogue ***3***

While the binding affinities of the open-chain Ipom-F analogues (**1**: −4.25 ± 0.6 kcal/mol; **2**: −4.45 ± 0.5 kcal/mol; **5**: −3.82 ± 0.6 kcal/mol) were predicted to be less favorable than those of the closed-chain Ipom-F (−7.82 ± 0.2 kcal/mol) and **22** (−8.09 ± 0.4 kcal/mol), it should be noted that it is difficult to compare compound docking affinities due to the different entropy changes between the open- and closed-chain analogues. Empirical docking scoring functions such as that used in Autodock 4.2 (Available online: https://autodocksuite.scripps.edu/autodock4/ (accessed on 2 June 2022)) typically approximate entropy penalties on binding using the number of rotatable bonds present in the ligand. Here, the difference in the docking score (~3 kcal/mol) is associated with the number of rotatable bonds present in the open-chain analogues **1**, **2** and **5** versus the closed-chain Ipom-F and **22** (respectively 31–32 rotatable bonds versus 15). In the present case, we, therefore, suggest that the docking score or binding affinity obtained from the docking software should not be considered as an accurate parameter to represent the experimental binding affinity. Adaptive biasing force (ABF)/metadynamics simulations may be helpful in future work to characterize the enthalpic and/or entropic contributions involved in the binding of closed-ring vs. open-ring Ipom-F derivatives.

Despite these caveats, we postulate that the disaccharide core is capable of controlling the overall conformation of ring-opened analogues that enables them to retain sufficient interaction with Sec61α. Therefore, we exploited our biology-directed and in silico-aided studies to design a new Ipom-F-derived lead compound that would be more synthetically accessible than compounds **1** and **2**, while also retaining a comparable level of biological activity. Since the alkene-reduced analogue **2** was the most potent open-chain analogue discovered to date, we first decided to remove both terminal double bonds from the new analogue. Second, and to avoid low-yielding Grignard reactions during the synthesis of the aglycones at the 6”-OH-Glu*p* and C-1′-Fuc*p* positions [8], we sought to replace 4-oxononanoic acid with mono-butyl succinate **28** (Figure 1) in the synthetic route to the new analogue. Such a strategy, while synthetically attractive, additionally permits the well-tolerated bioisosteric replacement of the C-5 methylene with an oxygen atom (cf. **22**; Table 1) while retaining the C-4 carbonyl group, whose removal is detrimental to compound potency (cf. **6**; Table 1).

Lastly, we decided to increase the lipophilicity of the new analogue for two reasons: (i) our earlier studies on how the ring size of closed-chain analogues affects compound potency revealed that ring expansion by two methylene units, and the concomitant increase in lipophilicity, is an advantageous feature [14]; and (ii) in this study, increased compound potency in vitro and in cellula for both the open-chain (**2** ≥ **5** > **1**) and closed-chain (Ipom-F > **22**) compounds appears to correlate with an increase in lipophilicity (based on cLogP calculations; Table 1).

We, therefore, considered the possibility that the chiral starting material (*S*)-4-hydroxy-1-nonene (which requires a three-step synthesis from an expensive, chiral reagent) could be replaced with a simpler and cheaper alternative; the achiral, but two extra methylene unit-containing, 6-undecanol. While uncertain about how loss of the 11*S* chiral center may impact compound potency, we postulated that the increased lipophilicity from the extra two methylene units may compensate for the likely significant loss in activity following its removal [8].

To this end, and before embarking on its synthesis, we sought to use molecular docking to evaluate the potential interactions of the stereochemically simplified and synthetically more accessible open-chain analogue **3** in the inhibited state of the Sec61α channel pore (cf. Figure 3 and Appendix A). Due to the limitation on the number of rotatable bonds that can be considered (maximum of 32 in AutoDock4.2), we docked the closely related compounds **3a** and **3b** instead (Figure 5A; respectively 32 and 31 rotatable bonds, with and without the 11*S* chiral center). Similar to the open-chain analogue 2, both analogues are predicted to bind both the mycolactone binding site and the middle of the lateral gate (Figure 5B). When positioned at the groove of mycolactone, **3a** and **3b** preferentially interact with the TM2, TM7, CL4 and CL8 regions of Sec61α (Appendix A) with predicted binding affinities of −4.15 ± 0.6 kcal/mol and −4.78 ± 0.6 kcal/mol, respectively. Similar to Ipom-F and analogue **2**, **3b** preferentially forms hydrogen bonds with Gln170 and Lys171 in the CL4 region and a hydrogen bond was identified in four out of five independent docking simulations. In contrast to **3b**, hydrogen bonding between **3a** and the CL4 region (Gln170 and Lys171) was only observed in two out of five independent docking simulations. This suggests that **3b** interacts more strongly with CL4 than **3a**.

When bound in the middle of the lateral gate, both analogues are predicted to be in contact with the plug region, TM2, TM3 and TM7 (Appendix A) with binding affinities of −4.7 ± 0.2 kcal/mol and −5.26 ± 0.9 kcal/mol, respectively. Furthermore, both analogues form hydrogen bonds (observed in four out of five independent docking simulations) with Thr86 (TM2) and Gln127 (TM3) when bound at the lateral gate. In that position, the open-chain analogue **2** also likely forms a hydrogen bond with Thr86 (TM2), while this bond was not observed in any docking simulation for the closed-chain analogue **22**.

Taken together, our molecular docking studies of **3a** and **3b** suggested that removal of the 11*S* chiral center may result in a greater number of contact residues with the CL4 region (62% and 77% respectively) and an increased potential to hydrogen bond with residues Gln170 and Lys171 (Appendix A). Thus, we elected to remove the 11*S* center, increase the lipophilicity of the fatty acid region and bioisosterically replace the C-5 methylene with an oxygen atom in our newly designed open-chain analogue **3**.

### 2.6. Synthesis of Analogue ***3***

Our synthesis of analogue **3** (Figure 1) started with the known trichloroacetimidate glucosyl donor **23** [14] and the diol fucoside acceptor **24** (see Appendix A). Regioselective glycosylation on 2-OH-Fuc*p* afforded the monohydroxy intermediate **25** in moderate yield and, after acetylation of the free 4′-OH-Fuc*p* in **25** with acetic anhydride, the isopropylidene protecting group was removed by trifluoroacetic acid to give 4″,6″-diol **27**. EDC-mediated Steglich esterification of 6″-OH-Glu*p* with mono-butyl succinate **28** (see Appendix A) successfully produced a second monohydroxy intermediate **29**, despite somewhat poor regioselectivity. The cinnamate moiety was then introduced to 4-OH-Glu*p* by the Mukaiyama reagent, 2-chloro-1-methylpyridinium iodide (CMPI), to give the intermediate **30** in excellent yield. Subsequently, both levulinoyl (Lev) groups were removed by hydrazine under buffered conditions [14,41]. In the penultimate step, the tiglate moiety was selectively introduced to 3″-OH-Glu*p* in **31** using CMPI to give the intermediate **32** in good yield. Finally, the TBS (*tert*-butyldimethylsilyl) group at 3′-OH-Fuc*p* was cleaved using TBAF and acetic acid in THF to give the target molecule **3**. In brief, **3** was synthesized in 7.5% yield (not optimized) over eight steps from two key monosaccharide building blocks **23** and **24**.

### 2.7. Analogue ***3*** Inhibits Sec61-Mediated Protein Translocation with Potency and Selectivity Comparable to Ipom-F and ***2***

Following the successful synthesis of analogue **3**, we first analyzed its effects on the in vitro membrane integration of our two model protein substrates, Ii and GypC (Figure 6A–C). In the first instance, 1 µM analogue **3** inhibited the membrane integration of Ii (Figure 6A, lane 4 versus lane 1) to a level comparable to that of Ipom-F, **2** and **5** (circa ~73–86% reduction in Ii membrane integration; Figure 3B,C). In the second instance, the same concentration of analogue **3** did not affect the membrane integration of GypC (Figure 6A, lanes 13–14 and Figure 6B), confirming the selective inhibition of Sec61-mediated protein translocation by analogue **3** in ER-derived microsomes. Thus, we proceeded to determine the estimated IC_50_ of analogue **3** on the in vitro membrane integration of Ii (Figure 6A, lanes 1–12, Figure 3D and Appendix A), which allowed us to rank order analogue **3** as the third most potent compound of the seven tested in this study: Ipom-F (IC_50_: 155 nM) > **2** (IC_50_: 202 nM) > **3** (IC_50_: 242 nM) > **5** (IC_50_: 291 nM) > **22** (IC_50_: 334 nM) > **1** (IC_50_: 562 nM) > **6** (negligible inhibition at 1 µM).

When we analyzed analogue **3** in our resazurin-based cell viability assay (cf. Figure 3F) using HCT116 Sec61-WT cells (Figure 6D,E and Appendix A), we found it to be the second most cytotoxic compound of the six that we performed IC_50_ analyses for: Ipom-F (IC_50_: 18 nM) > **3** (IC_50_: 40 nM) > **2** (IC_50_: 41 nM) = **5** (IC_50_: 41 nM) > **22** (IC_50_: 70 nM) > **1** (IC_50_: 170 nM). Furthermore, and as anticipated, cell death was not observed in resistance-conferring HCT116 Sec61-G80W mutant cells treated with the same concentration of analogue **3** that induced ~58% cell death in HCT116 Sec61-WT cells (Figure 6D). We, therefore, conclude that our chosen combination of advantageous chemical features permits the macrocyclic ring opening and removal of the 11*S* chiral center of Ipom-F without a significant loss in the potency or selectivity of the inhibition of Sec61-mediated protein translocation at the ER.

## 3. Materials and Methods

### 3.1. Chemical Synthesis General Methods

All reaction glassware was thoroughly washed and oven dried before any reactions were undertaken. Unless otherwise stated, all commercially obtained reagents were used without further purification and all reactions were conducted under argon atmosphere. Reaction progress was monitored by TLC using silica gel MF254 glass back plates with detection under UV lamp (254 nm) or charring with 5 % (*v*/*v*) H_2_SO_4_ (sulfuric acid) in EtOH (ethanol). Column chromatographic purifications were performed using silica gel (70–230 mesh) with a ratio that spanned from 100 to 50: 1 (*w*/*w*) between the silica gel and crude products. All ^1^H NMR spectra were obtained in deuterated chloroform (CDCl_3_), deuterated methanol (CD_3_OD) or deuterated dimethyl sulfoxide ((CD_3_)_2_SO using chloroform (CHCl_3_, *δ* = 7.27), methanol (CH_3_OH, *δ* = 3.31) or dimethyl sulfoxide ((CH_3_)_2_SO, *δ* = 2.50) as an internal reference for ^1^H. All ^13^C NMR spectra were proton decoupled and obtained in CDCl_3_, CD_3_OD or (CD_3_)_2_SO with CHCl_3_ (*δ* = 77.0), CH_3_OH (*δ* = 49.9) or CH_3_)_2_SO (*δ* = 40.4) as internal references for ^13^C. NMR data are reported in the form: chemical shifts (*δ*) in ppm, multiplicity, coupling constants (*J*) in Hz, and integrations. ^1^H data are reported as though they were first order. An error less than 0.5 Hz is reported for coupling constants between two coupled protons. Other 1D and 2D NMR spectra, such as ^135^DEPT, COSY, HMQC and HMBC, were collected in addition to ^1^H and ^13^C in the characterization of new compounds. Low-resolution mass spectra (LRMS) were obtained on a LTQ XL mass spectrometer equipped with an electrospray ion source using the positive ion mode and connected to a linear ion trap mass analyzer. Purity was analyzed using a Waters HPLC with a photodiode array (PDA) detector, a DIONEX Acclaim^®^ 120 reverse phase column (C18, 5 μm, 120Å, 4.6 × 150 mm) and an isocratic mobile phase of 83% acetonitrile in water at a flow rate of 1.5 mL/min.

#### 3.1.1. Synthesis of Compound **25**

The fucoside diol acceptor **24** (339.9 mg, 0.79 mmol), known glucoside trichloroacetimidate donor **23** [14] (478.1 mg, 0.85 mmol, 1.08 eq.) and crushed activated 4Å molecular sieves (1 g) were suspended in anhydrous CH_2_Cl_2_ (7 mL). The mixture was stirred under an argon atmosphere for ~30 min at room temperature and then cooled to −78 °C. TMSOTf (14.2 μL, 0.079 mmol, 0.1 eq.) was added dropwise via a syringe and the reaction mixture was gradually warmed to −20 °C over ~1 h. The reaction mixture was then quenched by the addition of Et_3_N and filtered through a pad of celite and the filtrate concentrated. The resulting residue was purified by column chromatography (15:1→6:1 hexanes–EtOAc) to acquire pure **25** as a colorless oil (315.1 mg, 49%): R*_f_* 0.56 (4:1 hexanes–EtOAc); ^1^H NMR (400 MHz, CDCl_3_, δ_H_) 5.10 (d, *J* = 7.8 Hz, 1H), 5.03 (t, *J* = 9.4 Hz, 1H), 4.83 (dd, *J* = 10.4 Hz, *J* = 7.9 Hz, 1H), 4.21 (d, *J* = 7.5 Hz, 1H), 3.91 (dd, *J* = 10.9 Hz, *J* = 5.3 Hz, 1H), 3.65–3.79 (m, 3H), 3.62 (t, *J* = 9.6 Hz, 1H), 3.42–3.57 (m, 3H), 3.21 (qd, *J* = 9.7 Hz, *J* = 5.5 Hz, 1H), 2.45–2.81 (m, 9H), 2.13 (s, 6H), 1.10–1.57 (m, 25H), 0.80–0.95 (m, 15H), 0.12 (s, 3H), 0.07 (s, 3H); ^13^C NMR (100 MHz, CDCl_3_, δ_C_) 206.5 (C=O), 206.4 (C=O), 172.1 (C=O), 171.3 (C=O), 101.1 (O_2_CH), 99.8 (O_2_C), 99.6 (O_2_CH), 79.8 (OCH), 76.0 (OCH), 75.6 (OCH), 73.9 (OCH), 72.8 (OCH), 72.7 (OCH), 71.5 (OCH), 69.6 (OCH), 67.5 (OCH), 62.3 (OCH_2_), 37.8 (CH_2_), 37.7 (CH_2_), 34.5 (CH_2_), 33.7 (CH_2_), 32.3 (CH_2_), 32.1 (CH_2_), 29.9(0) (CH_3_), 29.8(9) (CH_3_), 29.0 (CH_3_), 28.0 (CH_2_), 27.9 (CH_2_), 26.1 (C(CH_3_)_3_), 24.9 (CH_2_), 24.6 (CH_2_), 22.9 (CH_2_), 22.7 (CH_2_), 18.9 (CH_3_), 18.1 (SiC(CH_3_)_3_), 16.5 (CH_3_), 14.3 (CH_3_), 14.2 (CH_3_), −4.1 (SiCH_3_), −4.5 (SiCH_3_).

#### 3.1.2. Synthesis of Compound **26**

To an ice-cold solution of **25** (315.1 mg, 0.38 mmol), DMAP (4.6 mg, 0.1 eq.) and Et_3_N (157.6 μL, 1.14 mmol, 3.0 eq.) in DCM (3 mL), Ac_2_O was added (44.8 μL, 0.47 mmol, 1.25 eq.). The mixture was warmed to room temperature overnight and the reaction was then quenched by the addition of methanol. The resulting solution was extracted with DCM and the organic fractions washed with 1N HCl and saturated NaHCO_3_. The collected organic layer was dried over Na_2_SO_4_, filtered and concentrated under vacuum to yield the crude product as a pale yellow syrup (288.3 mg), which was used directly for the next step without further purification.

#### 3.1.3. Synthesis of Compound **27**

To a solution of **26** (288.3 mg, 0.33 mmol) in CHCl_3_ (5 mL) trifluoroacetic acid (TFA) was added (126.9 μL, 1.65 mmol, 5.0 eq.). The mixture was stirred at room temperature for 4 h and then quenched by Et_3_N. After evaporation, the residue was purified by column chromatography (8:1→2:1, hexanes–EtOAc) to afford **27** as a colorless syrup (217.6 mg, 69% over two steps): R*_f_* 0.43 (2:1 hexanes–EtOAc); ^1^H NMR (400 MHz, CDCl_3_, δ_H_) 4.92–5.02 (m, 3H), 4.83 (dd, *J* = 9.7 Hz, *J* = 7.8 Hz, 1H), 4.20 (d, *J* = 7.7 Hz, 1H), 3.79–3.93 (m, 2H), 3.62–3.79 (m, 3H), 3.43–3.61 (m, 3H), 3.32–3.40 (m, 1H), 2.68–2.80 (m, 3H), 2.41–2.68 (m, 6H), 2.13 (s, 3H), 2.12 (s, 3H), 2.09 (s, 3H), 1.13–1.61 (m, 16H), 1.07 (d, *J* = 6.4 Hz, 3H), 0.75–0.94 (m, 15H), 0.09 (s, 3H), 0.05 (s, 3H); ^13^C NMR (100 MHz, CDCl_3_, δ_C_) 208.3 (C=O), 206.3 (C=O), 173.1 (C=O), 171.4 (C=O), 171.0 (C=O), 101.5 (O_2_CH), 98.8 (O_2_CH), 81.5 (OCH), 76.4 (OCH), 75.0 (OCH), 74.9 (OCH), 73.7 (OCH), 73.4 (OCH), 72.3 (OCH), 69.8 (OCH), 68.8 (OCH), 62.1 (OCH_2_), 38.4 (CH_2_), 37.8 (CH_2_), 34.6 (CH_2_), 34.0 (CH_2_), 32.2 (CH_2_), 32.0 (CH_2_), 29.8(8) (CH_3_), 29.8(5) (CH_3_), 28.1 (CH_2_), 28.0 (CH_2_), 25.9 (C(CH_3_)_3_), 24.8 (2xCH_2_), 22.8 (CH_2_), 22.7 (CH_2_), 21.1 (CH_3_), 17.8 (SiC(CH_3_)_3_), 16.6 (CH_3_), 14.3 (CH_3_), 14.2 (CH_3_), −4.2 (SiCH_3_), −4.5 (SiCH_3_).

#### 3.1.4. Synthesis of Compound **29**

To an ice-cold solution of **27** (217.6 mg, 0.26 mmol), EDC (125.1 mg, 0.65 mmol, 2.5 eq.) and DMAP (8.0 mg, 0.065 mmol, 0.25 eq.) in DCM (7 mL), **28** was added (52.5 mg, 0.30 mmol, 1.16 eq.), and the reaction was warmed to room temperature overnight. The solution was then washed with 1N HCl and saturated NaHCO_3_ and the collected organic fractions were dried over Na_2_SO_4_, filtered and concentrated under vacuum. The resulting residue was then purified by column chromatography (8:1→2:1, hexanes–EtOAc) to afford **29** as a colorless syrup (104.0 mg, 40%): R*_f_* 0.52 (1:1 hexanes–EtOAc); ^1^H NMR (400 MHz, CDCl_3_, δ_H_) 5.03 (d, *J* = 7.8 Hz, 1H), 4.91–5.01 (m, 2H), 4.83 (dd, *J* = 9.7 Hz, *J* = 7.8 Hz, 1H), 4.41 (dd, *J* = 8.1 Hz, *J* = 4.2 Hz, 1H), 4.29 (dd, *J* = 12.0 Hz, *J* = 2.3 Hz, 1H), 4.23 (d, *J* = 7.6 Hz, 1H), 4.04 (t, *J* = 6.7 Hz, 2H), 3.85 (dd, *J* = 9.1 Hz, *J* = 7.6 Hz, 1H), 3.76 (dd, *J* = 9.4 Hz, *J* = 3.5 Hz, 1H), 3.40–3.66 (m, 5H), 2.41–2.85 (m, 12H), 2.13 (s, 3H), 2.11 (s, 3H), 2.07 (s, 3H), 1.14–1.64 (m, 20H), 1.07 (d, *J* = 6.4 Hz, 3H), 0.77–0.96 (m, 18H), 0.09 (s, 3H), 0.05 (s, 3H); ^13^C NMR (100 MHz, CDCl_3_, δ_C_) 207.9 (C=O), 206.3 (C=O), 173.0 (C=O), 172.6 (C=O), 172.5 (C=O), 171.3 (C=O), 170.9 (C=O), 101.4 (O_2_CH), 98.9 (O_2_CH), 81.0 (OCH), 76.0 (OCH), 75.1 (OCH), 73.7(5) (OCH), 73.7(0) (OCH), 73.6(6) (OCH), 72.5 (OCH), 69.1 (OCH), 68.8 (OCH), 64.8 (OCH_2_), 63.3 (OCH_2_), 38.3 (CH_2_), 37.8 (CH_2_), 34.5 (CH_2_), 33.9 (CH_2_), 32.2 (CH_2_), 32.1 (CH_2_), 30.6 (CH_2_), 29.9 (CH_3_), 29.8 (CH_3_), 29.2 (CH_2_), 29.0 (CH_2_), 28.1 (CH_2_), 28.0 (CH_2_), 25.9 (C(CH_3_)_3_), 24.8 (CH_2_), 24.7 (CH_2_), 22.9 (CH_2_), 22.7 (CH_2_), 21.0 (CH_3_), 19.1 (CH_2_), 17.8 (SiC(CH_3_)_3_), 16.6 (CH_3_), 14.3 (CH_3_), 14.2 (CH_3_), 13.8 (CH_3_), −4.3 (2xSiCH_3_).

#### 3.1.5. Synthesis of Compound **30**

To an ice-cold solution of **29** (104.0 mg, 0.105 mmol), cinnamic acid (23.4 mg, 0.158 mmol, 1.5 eq.), CMPI (53.7 mg, 0.210 mmol, 2.0 eq.) and DMAP (12.8 mg, 0.105 mmol, 1.0 eq.) in DCM (4 mL), Et_3_N was added (72.9 μL, 0.525 mmol, 5.0 eq.). The mixture was warmed to room temperature overnight. After evaporation, the resulting residue was then purified by column chromatography (10:1→4:1, hexanes–EtOAc) to afford **30** as a colorless syrup (108.1 mg, 92%): R*_f_* 0.46 (2:1 hexanes–EtOAc); ^1^H NMR (400 MHz, CDCl_3_, δ_H_) 7.62 (d, *J* = 16.0 Hz, 1H), 7.43–7.53 (m, 2H), 7.29–7.38 (m, 3H), 6.33 (d, *J* = 16.0 Hz, 1H), 5.06–5.25 (m, 3H), 4.91–5.00 (m, 2H), 4.20–4.28 (m, 2H), 4.16 (dd, *J* = 12.1 Hz, *J* = 2.9 Hz, 1H), 4.01 (t, *J* = 6.7 Hz, 2H), 3.89 (dd, *J* = 9.3 Hz, *J* = 7.6 Hz, 1H), 3.80 (dd, *J* = 9.4 Hz, *J* = 3.5 Hz, 1H), 3.45–3.70 (m, 3H), 2.35–2.80 (m, 12H), 2.12 (s, 3H), 2.08 (s, 3H), 2.03 (s, 3H), 1.15–1.61 (m, 20H), 1.08 (d, *J* = 6.4 Hz, 3H), 0.80–0.98 (m, 18H), 0.13 (s, 3H), 0.08 (s, 3H); ^13^C NMR (100 MHz, CDCl_3_, δ_C_) 206.4 (C=O), 206.2 (C=O), 172.2 (C=O), 172.0 (2xC=O), 171.3 (C=O), 170.9 (C=O), 165.4 (C=O), 146.5 (=CH), 134.1 (=C), 130.7 (=CH), 129.0 (2x=CH), 128.4 (2x=CH), 116.7 (=CH), 101.4 (O_2_CH), 98.9 (O_2_CH), 81.3 (OCH), 75.1 (OCH), 73.8 (OCH), 73.6 (OCH), 73.0 (OCH), 72.5 (OCH), 71.8 (OCH), 68.9 (OCH), 68.8 (OCH), 64.6 (OCH_2_), 62.8 (OCH_2_), 37.8(3) (CH_2_), 37.7(6) (CH_2_), 34.5 (CH_2_), 34.0 (CH_2_), 32.3 (CH_2_), 32.1 (CH_2_), 30.6 (CH_2_), 29.9 (CH_3_), 29.6 (CH_3_), 29.1 (CH_2_), 29.0 (CH_2_), 28.0 (CH_2_), 27.9 (CH_2_), 26.0 (C(CH_3_)_3_), 25.0 (CH_2_), 24.7 (CH_2_), 23.0 (CH_2_), 22.7 (CH_2_), 21.0 (CH_3_), 19.1 (CH_2_), 17.8 (SiC(CH_3_)_3_), 16.6 (CH_3_), 14.3 (CH_3_), 14.2 (CH_3_), 13.8 (CH_3_), −4.3 (SiCH_3_), −4.4 (SiCH_3_).

#### 3.1.6. Synthesis of Compound **31**

To an ice-cold solution of **30** (108.1 mg, 0.0966 mmol) in DCM (3 mL), a buffer solution of hydrazine monohydrate was added (25.8 μL, 0.532 mmol, 5.5 eq.) in acetic acid (354.1 μL) and pyridine (531.1 μL). The mixture was warmed to room temperature and stirred for 4 h, then quenched by acetone, diluted with DCM and washed with brine. The organic layer was dried over Na_2_SO_4_, filtered and concentrated under vacuum. The resulting residue was subsequently purified by column chromatography (10:1→4:1, hexanes–EtOAc) to afford **31** as a colorless syrup (88.3 mg, 99%): R*_f_* 0.57 (2:1 hexanes–EtOAc); ^1^H NMR (400 MHz, CDCl_3_, δ_H_) 7.69 (d, *J* = 16.0 Hz, 1H), 7.45–7.55 (m, 2H), 7.30–7.40 (m, 3H), 6.41 (d, *J* = 16.0 Hz, 1H), 5.12 (t, *J* = 9.0 Hz, 1H), 5.05 (br s, 1H), 4.60 (d, *J* = 7.7 Hz, 1H), 4.39 (d, *J* = 7.4 Hz, 1H), 4.26 (dd, *J* = 11.3 Hz, *J* = 3.7 Hz, 1H), 4.14 (br d, *J* = 12.0 Hz, 1H), 4.03 (t, *J* = 6.7 Hz, 2H), 3.85–3.94 (m, 2H), 3.55–3.84 (m, 5H), 3.44 (t, *J* = 8.0 Hz, 1H), 2.50–2.73 (m, 5H), 2.10 (s, 3H), 1.17–1.68 (m, 20H), 1.12 (d, *J* = 6.4 Hz, 3H), 0.80–0.95 (m, 18H), 0.17 (s, 3H), 0.13 (s, 3H); ^13^C NMR (100 MHz, CDCl_3_, δ_C_) 172.3 (C=O), 172.2 (C=O), 170.8 (C=O), 166.1 (C=O), 146.4 (=CH), 134.2 (=C), 130.7 (=CH), 129.0 (2x=CH), 128.3 (2x=CH), 117.1 (=CH), 103.6 (O_2_CH), 101.2 (O_2_CH), 79.9 (OCH), 78.7 (OCH), 76.0 (OCH), 73.7 (OCH), 73.1 (OCH), 72.9 (OCH), 72.8 (OCH), 70.1 (OCH), 68.8 (OCH), 64.6 (OCH_2_), 62.8 (OCH_2_), 34.1 (CH_2_), 33.4 (CH_2_), 32.2 (CH_2_), 32.0 (CH_2_), 30.7 (CH_2_), 29.1 (CH_2_), 29.0 (CH_2_), 26.0 (C(CH_3_)_3_), 25.0 (CH_2_), 24.4 (CH_2_), 22.7 (2xCH_2_), 20.9 (CH_3_), 19.2 (CH_2_), 18.0 (SiC(CH_3_)_3_), 16.6 (CH_3_), 14.2(0) (CH_3_), 14.1(8) (CH_3_), 13.8 (CH_3_), −4.2 (SiCH_3_), −4.5 (SiCH_3_).

#### 3.1.7. Synthesis of Compound **32**

To an ice-cold solution of **31** (96.0 mg, 0.104 mmol), tiglic acid (15.5 mg, 0.155 mmol, 1.5 eq.), CMPI (105.7 mg, 0.414 mmol, 4.0 eq.) and DMAP (12.7 mg, 0.104 mmol, 1.0 eq.) in DCM (2 mL), Et_3_N was added (72.1 μL, 0.520 mmol, 5.0 eq.). The mixture was warmed to room temperature overnight. After evaporation, the resulting residue was then purified by column chromatography (10:1→4:1, hexanes–EtOAc) to afford **32** as a colorless syrup (91.0 mg, 87%): R*_f_* 0.55 (2:1 hexanes–EtOAc); ^1^H NMR (400 MHz, CDCl_3_, δ_H_) 7.59 (d, *J* = 16.0 Hz, 1H), 7.42–7.51 (m, 2H), 7.30–7.39 (m, 3H), 6.74–6.85 (m, 1H), 6.30 (d, *J* = 16.0 Hz, 1H), 5.20–5.34 (m, 2H), 5.04 (d, *J* = 3.3 Hz, 1H), 4.73 (d, *J* = 7.7 Hz, 1H), 4.39 (d, *J* = 7.6 Hz, 1H), 4.26 (dd, *J* = 11.3 Hz, *J* = 4.1 Hz, 1H), 4.14 (dd, *J* = 12.3 Hz, *J* = 2.3 Hz, 1H), 4.03 (t, *J* = 6.7 Hz, 2H), 3.92 (dd, *J* = 9.7 Hz, *J* = 7.6 Hz, 1H), 3.72–3.83 (m, 2H), 3.54–3.72 (m, 4H), 2.50–2.70 (m, 4H), 2.07 (s, 3H), 1.65–1.76 (m, 6H), 1.16–1.65 (m, 20H), 1.10 (d, *J* = 6.4 Hz, 3H), 0.80–0.95 (m, 18H), 0.17 (s, 3H), 0.12 (s, 3H); ^13^C NMR (100 MHz, CDCl_3_, δ_C_) 172.2 (C=O), 172.1 (C=O), 170.9 (C=O), 167.6 (C=O), 165.5 (C=O), 146.3 (=CH), 138.3 (=CH), 134.1 (=C), 130.7 (=CH), 129.0 (2x=CH), 128.3 (2x=CH), 128.0 (=C), 116.8 (=CH), 103.9 (O_2_CH), 101.1 (O_2_CH), 79.6 (OCH), 78.7 (OCH), 74.4 (OCH), 73.2 (OCH), 72.9 (OCH), 72.7(7) (OCH), 72.7(6) (OCH), 68.8 (OCH), 68.4 (OCH), 64.6 (OCH_2_), 62.7 (OCH_2_), 34.0 (CH_2_), 33.4 (CH_2_), 32.2 (CH_2_), 32.0 (CH_2_), 30.7 (CH_2_), 29.1 (CH_2_), 29.0 (CH_2_), 25.9 (C(CH_3_)_3_), 25.0 (CH_2_), 24.4 (CH_2_), 22.8 (CH_2_), 22.7 (CH_2_), 21.0 (CH_3_), 19.1 (CH_2_), 17.9 (SiC(CH_3_)_3_), 16.6 (CH_3_), 14.5 (CH_3_), 14.2(1) (CH_3_), 14.1(9) (CH_3_), 13.8 (CH_3_), 12.1 (CH_3_), −4.3 (SiCH_3_), −4.5 (SiCH_3_).

#### 3.1.8. Synthesis of Analogue **3**

To a solution of **32** (83.4 mg, 0.083 mmol) and acetic acid (190 μL, 3.32 mmol, 40 eq.) in THF (3 mL), tetra-*n*-butylammonium fluoride (TBAF) solution was added in THF (1.0 M, 1.66 mL, 1.66 mmol, 20 eq.) at room temperature and the mixture was stirred overnight. The solution was diluted with CH_2_Cl_2_ and successively washed with 1N HCl, saturated aqueous NaHCO_3_ and brine. The collected organic layer was then dried over Na_2_SO_4_ and filtered. The filtrate was concentrated under vacuum and the resulting residue purified by column chromatography (8:1→2:1, hexanes–EtOAc) to afford **3** as a colorless to pale yellow syrup (51.7 mg, 70%): R*_f_* 0.31 (2:1 hexanes–EtOAc); ^1^H NMR (400 MHz, CDCl_3_, δ_H_) 7.62 (d, *J* = 16.0 Hz, 1H), 7.43–7.54 (m, 2H), 7.30–7.42 (m, 3H), 6.79–6.91 (m, 1H), 6.31 (d, *J* = 16.0 Hz, 1H), 5.13–5.31 (m, 3H), 4.70 (d, *J* = 8.2 Hz, 1H), 4.38 (d, *J* = 7.3 Hz, 1H), 4.31 (br s, 1H), 4.15–4.26 (m, 2H), 4.04 (t, *J* = 6.7 Hz, 2H), 3.62–3.89 (m, 7H), 2.50–2.69 (m, 4H), 2.18 (s, 3H), 1.66–1.77 (m, 6H), 1.18–1.61 (m, 20H), 1.16 (d, *J* = 6.4 Hz, 3H), 0.80–0.95 (m, 9H); ^13^C NMR (100 MHz, CDCl_3_, δ_C_) 172.3 (C=O), 172.1 (C=O), 171.4 (C=O), 168.0 (C=O), 165.6 (C=O), 146.6 (=CH), 139.1 (=CH), 134.0 (=C), 130.8 (=CH), 129s.0 (2x=CH), 128.4 (2x=CH), 127.8 (=C), 116.5 (=CH), 102.7 (O_2_CH), 99.6 (O_2_CH), 79.3 (OCH), 77.6 (OCH), 74.3 (OCH), 72.8 (OCH), 72.2 (OCH), 71.8 (OCH), 70.8 (OCH), 69.3 (OCH), 68.3 (OCH), 64.7 (OCH_2_), 62.6 (OCH_2_), 34.4 (CH_2_), 33.4 (CH_2_), 32.0 (CH_2_), 31.9 (CH_2_), 30.7 (CH_2_), 29.1 (CH_2_), 29.0 (CH_2_), 24.8 (2xCH_2_), 22.7 (2xCH_2_), 21.1 (CH_3_), 19.2 (CH_2_), 16.3 (CH_3_), 14.6 (CH_3_), 14.1(9) (CH_3_), 14.1(5) (CH_3_), 13.8 (CH_3_), 12.1 (CH_3_). LRMS (ESI) *m*/*z* calcd for C_47_H_70_NaO_16_ [M+Na]^+^: 913. Found: 913. Purity: 97.1% (MeCN/H_2_O 83:17; 1.5 mL/min, *t*_R_ = 14.9 min, Appendix A).

### 3.2. Biological Analysis

All compounds from stock solutions in DMSO, or an equivalent volume of DMSO, were included at 5% (*v*/*v*) in membrane insertion assays or 20% (*v*/*v*) in cytotoxicity assays.

#### 3.2.1. In Vitro Membrane Insertion Assay

Linear DNA of the short form of human HLA class II histocompatibility antigen gamma chain (Ii; P04232, isoform 2, residues 17–232) or human glycophorin C (GypC; P04921) were generated by PCR and transcribed into RNA using T7 RNA polymerase (Promega). Membrane insertion assays (20 µL, 1 h at 30 °C, containing 6.5% (*v*/*v*) nuclease-treated ER microsomes (from stock with OD280 = 44/mL)), endoglycosidase H_f_ (New England Biolabs) treatment, sample resolution by SDS-PAGE (16% polyacrylamide gels) and gel drying were performed as previously described [13,14,37,38,42]. Following exposure to a phosphorimaging plate for 24–72 h, radiolabeled products were visualized using a Typhoon FLA-7000 (GE Healthcare, Tokyo, Japan) and the ratio of the signal intensity for the N-glycosylated (XGly) and non-glycosylated (0Gly) forms obtained using AIDA v.5.0 (Raytest Isotopenmeβgeräte). This value was then expressed relative to the matched DMSO control (set to 100%) in order to estimate the mean relative insertion (± SEM) from insertion experiments performed in triplicate (*n* = 3, biologically independent experiments). IC_50_ value estimates were determined in Prism 8 (GraphPad, San Diego, CA, USA) using nonlinear regression to fit data to a curve of variable slope (four parameters) using the least-squares fitting method, with the top and bottom plateaus of the curve defined as 100% and 7.67% (the mean of all data at 500 µM across all compounds), respectively [13,14].

#### 3.2.2. Cell Culture and Resazurin-Based Viability Assays

The human breast cancer cell line (MDA-MB-231) was maintained in a DMEM high glucose culture medium supplemented with 10% (*v*/*v*) fetal bovine serum (FBS) and 2 mM *L*-glutamine. Parental (Sec61α-WT) [13] HCT-116 (human colorectal cancer cells, ATCC, CCL-247) and mutant (Sec61α-G80W) [22] HCT-116 cells were maintained in McCoy’s 5A (modified) medium ((ThermoFisher, Waltham, MA, USA, 16600-082)) supplemented with 10% (*v*/*v*) FBS (Gibco, 10500-064) and 100 units/mL penicillin and 100 mg/mL streptomycin (Gibco, cat: 15140-122). All cell lines were maintained in a 5% CO_2_ humidified incubator at 37 °C. Cytotoxicity assays were performed in triplicate sets as previously described [13,22] and viable cells were counted using an automated cell counter (Bio-Rad TC20) immediately before each experiment. Compound stock solutions in DMSO (10 mM) were diluted with supplemented culture media (MDA-MB-231 cells: high glucose DMEM; HCT116 Sec61α-WT and Sec61α-G80W cells: McCoy’s (modified) 5A) to make a series of gradient fresh working solutions at equal DMSO percentage immediately prior to each test. First, 100 mL of cells at a cell density of 5 × 10^4^ cells/mL (MDA-MB-231 cells) or 2.5 × 10^4^ cells/mL (Sec61α-WT or Sec61α-G80W HCT-116 cells) were seeded in black 96-well microtiter plates (Falcon, product 353219; 2500 cells/well) and incubated at 37° C for 24 h. Subsequently, the cells were treated with either 100 mL of the freshly made gradient working solution in a total volume of 200 mL/well (MDA-MB-231 cells) or 25 mL of freshly made gradient in a total volume of 125 mL/well (Sec61α-WT or Sec61α-G80W HCT-116 cells) at 37 °C for 72 h. For MDA-MB-231 cells, the media were discarded and 200 mL fresh medium containing 10% (*v*/*v*) alamarBlue HS cell viability reagent (resazurin stock solution) (ThermoFisher, Waltham, MA, USA, A50100) was added to each well. For Sec61α-WT or Sec61α-G80W HCT-116 cells, the media was not discarded and alamarBlue HS cell viability reagent (resazurin stock solution) (ThermoFisher, Waltham, MA, USA, A50100) was added to 10% (*v*/*v*). After that, all cells were incubated at 37 °C for a further 1–3 h and the emission of each well at 620 nm was detected using a Synergy H1 Hybrid multi-mode plate reader (BioTek, Agilent Technologies, Palo Alto, CA, USA) at excitation 580 nm. The percentage viability compared to the negative control (DMSO-treated cells) was determined and Prism 6 or 8 (GraphPad, San Diego, CA, USA) used to make a plot of viability (%) versus sample concentration and to calculate the concentration at which each compound exhibited 50% cytotoxicity (IC_50_). IC_50_ value estimates were determined using nonlinear regression to fit data to a curve of variable slope (four parameters) using the least-squares fitting method. For HCT-116 cell IC_50_ curves, the top and bottom plateaus were defined as 100% and 23.70% (the mean of all data at 25 µM across all compounds), respectively.

#### 3.2.3. Quantification and Statistical Analysis

Quantification procedures used in in vitro and in cellula experiments are described in Section 3.2.1 and Section 3.2.2. For all in vitro data, quantifications are given as means ± SEM for independent membrane insertion experiments performed in triplicate (*n* = 3, biologically independent experiments) and statistical significance with respect to DMSO controls (set as 100%) was determined using Tukey’s multiple comparison test (Figure 3C, two-way ANOVA) or unpaired t tests (Figure 6B, one-way ANOVA). For in cellula data using HCT116 Sec61α-WT cells, quantifications normalized to the DMSO control (set to 100%) are from one experiment (Figure 3G and Figure 6D left, *n* = 1) or given as means ± SEM from two (Figure 3H and Figure 6E, *n* = 2: Ipom-F, **22** and **3** (5–1 nM), **1**, **2** and **5** (1 nM)) or three (Figure 3H and Figure 6E, *n* = 3: Ipom-F, **22** and **3** (25 µM–25 nM), **1**, **2** and **5** (25 µM–5 nM)) independent resazurin-based cytotoxicity experiments. For all in cellula data using HCT116 Sec61α-G80W cells, quantifications normalized to the DMSO control (set to 100%) are given as means ± SEM from three independent resazurin-based cytotoxicity screens (*n* = 3). Statistical significance comparing the viability of HCT116 Sec61α-WT and HCT116 Sec61α-G80W cells was determined by ordinary one-way ANOVA and Dunnett’s multiple comparisons test (Figure 3G) or an unpaired t test (Figure 6E). In all cases, DF and F values are depicted in the appropriate figures and statistical significance is given as n.s., non-significant *p* > 0.1; *, *p* < 0.05; **, *p* < 0.01; ***, *p* < 0.001 and ****, *p* < 0.0001.

### 3.3. Homology Modeling and Docking Protocols

The 476 amino acid protein sequence of human Sec61α isoform 1 was retrieved from Uniprot (ID: P61619). The crystal structure of human Sec61α is not available; however, crystal structures of mammalian (canine) Sec61α have been reported [22]. Human Sec61α and Sec61α from *Canis lupus* (Uniprot ID: P38377) share 99.8% sequence identity. Hence, homology modeling was carried out to generate a three-dimensional conformational model of human Sec61α using a cryo-EM structure of the “inhibited state” of canine Sec61α as a template. Precisely, we used the cryo-EM structure reported for the inhibited state of canine Sec61α in the presence of mycolactone (6Z3T with resolution 2.6 Å) [8]. We added structural information for the missing part of 6Z3T by homology modeling based on 2WWB (EM structure with resolution 6.48 Å) [43]. The combined structure using 6Z3T and 2WWB was used as template for human Sec61α in homology modeling that was performed using MODELLER 9.21 [44]. After sequence alignment of target and template, MODELLER 9.21 was run locally with the automodel class to generate 50 different models. The model with the lowest DOPE score was selected as the final model and subjected to 1000 steps of energy minimization with the steepest descent algorithm, using the GROMACS (Available online: https://manual.gromacs.org/current/install-guide/index.html (accessed on 2 June 2022))package (version 5.0.7) [45] to relax side chain atoms. All compounds were modeled using structures drawn in ChemDraw Professional (CambridgeSoft, Waltham, MA, USA).

Docking of compounds was conducted using AutoDock4.2 [46] to predict energetically favorable binding poses of the compound inside or on the surface of human Sec61α. The docking calculations were performed in two consecutive steps. In the first docking step, we adopted a relatively large grid box (100 Å × 100 Å × 126 Å) covering the entire cavity of Sec61α, because the binding site(s) of these compounds are unknown. The Lamarckian genetic algorithm was employed with a population size of 150, 27 × 10^3^ generations and 25 × 10^5^ energy evaluations. All other docking parameters were set to the default values of AutoDock4.2. 1000 individual docking results were clustered according to a threshold for structural similarity of 2.0 Å RMSD. In each cluster, the representative conformation was set to the one with the lowest binding free energy for that cluster. Three independent sets of 1000 docking runs each were conducted in the first stage.

The first docking stage revealed that the compounds **22**, **2**, **3a** and **3b** dock favorably at two locations (the binding site of mycolactone and the lateral gate) within 1 kcal/mol. However, alternative poses for Ipom-F, **5** and **1** had predicted binding scores that are ~4, ~4 and ~2 kcal/mol less favorable than their best poses, respectively. Therefore, two small grid boxes are used at two different locations for compounds **22**, **2**, **3a** and **3b** in the second docking stage.

In the second docking stage, the size of the grid box was scaled down based on the population of the most stable binding positions of each compound. In this finer run, more stringent parameters were used; namely, cubic boxes of 86 Å × 86 Å × 70 Å, 0.5 × 10^6^ generations and 100 × 10^6^ energy evaluations. At this stage, we executed five independent fine docking runs yielding 50 docking results each. These 50 conformations were clustered similarly to those reported in a related docking study involving mycolactone [39]. The most favorable conformation of the Sec61α-analogue complex with lowest binding affinity score was selected for further analysis and considered as the final docking pose. Hydrogen bonding and contact residues (≤ 4Å) were identified by LigPlot+ [47] using default parameters.

### 3.4. Data and Software

In vitro data were analyzed with AIDA v5.0 ( Elysia-Raytest, Straubenhard, Germany). Homology modeling and docking analysis were performed with Modeller v9.24 and Autodock4.2, as described in the previous sections.

## 4. Conclusions

In summary, we conducted systematic studies on the first series of ring-opened analogues amongst all resin glycosides. We demonstrate that Ipom-F can be replaced with highly effective open-chain analogues that exert their cytotoxicity through the same molecular mechanism as their closed-chain counterparts; that is, by inhibiting Sec61-mediated protein translocation at the ER. The open-chain analogues **2** and **3** are defined as the most potent acyclic translocation inhibitors discovered to date. Thus, in contrast to coibamide A [35], opening of the Ipom-F macrocycle does not appear to result in a significant loss of either cytotoxicity or Sec61 inhibition (both in vitro and in cellula). We speculate that the disaccharide core provides the necessary [10] and sufficient conformational control so that the overall open-chain scaffold can still fit well into the binding pocket(s) of Sec61α. This is supported by our modeling studies, suggesting that these compounds can interact at one or more sites on the Sec61α subunit. Thus, we hypothesize that the flexibility of open-chain Ipom-F analogues may enhance their ability to perturb the normally stepwise binding of signal peptides to the Sec61 complex; a feature that may potentially be exploited towards the substrate-specific inhibition of Sec61-dependent protein clients in the future.

Synthetically, this acyclic structural framework allows us to bypass the two most challenging transition metal catalyzed reactions, namely ring-closing metathesis (RCM) and chemo-selective hydrogenation, which limit the production scale and flexibility of all current syntheses of Ipom-F [6,7,8]. Moreover, by incorporating the chemically advantageous features gleaned from our IC_50_ analyses of Ipom-F and analogues **1**, **2**, **5** and **22** (increased lipophilicity, expansion of the fatty acid portion, retention of the C-4 carbonyl group and bioisosteric replacement of the C-5 methylene with an oxygen atom; see also Table 1), we were able to synthesize open-chain analogues more efficiently by avoiding (i) a low-yielding Grignard reaction (10–30%) during the synthesis of the fatty acid fragment (4-oxo-8-nonenoic acid) at the 6”-OH-Glup position [8] and (ii) a three-step synthesis for the aglycone ((*S*)-4-hydroxy-1-nonene) at the C-1′-Fucp position through replacement of an expensive chiral starting reagent ((*S*)-(+)-epichlorohydrin) [8] with a commercially available, greener and non-chiral alternative (6-undecanol, cf. Figure 1). Therefore, for the first time, we were able to remove the natural 11*S* configuration on the fatty acid chain, which has proven crucial for the biological activity of Ipom-F [8] and is a universal feature for all resin glycosides. Taken together, we have revolutionized the synthesis of a potent Sec61 inhibitor in a more scalable and flexible manner than the parent Ipom-F and present **3** as a new and the most synthetically accessible lead compound.

To conclude, the work presented here ensures future ipomoeassin research using the ring-opened scaffold, which will help our efforts to explore and exploit the complete function of the Sec61 translocon for drug discovery. More broadly, our findings may be extended to other resin glycosides that could inspire exploration of new ring-opened analogues derived from this unique category of macrolactone natural products.

## Data Availability

Data are contained within the article or Appendix A.

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
