# Peer review of "Synthesis, Biological Evaluation and Docking Studies of Ring-Opened Analogues of Ipomoeassin F"

_molecules, 2022, doi:10.3390/molecules27144419_

Round 1

Reviewer 1 Report

The manuscript presents the study of open-chain analogues of Ipomeassin F, a macrocyclic resin glycoside exerting cytotoxic effects through inhibition of Sec61-mediated protein translocation at the ER. 

The work follows previous research by the authors on the mechanisms underlying Ipom-F-induced cytotoxicity and their medicinal applications. Here the authors validate that the integrity of the macrocyclic ring is not essential for bioactivity, as suggested by previous preliminary data [9, 13].

This finding is relevant and adds knowledge to the field as most natural Sec61 inhibitors feature a cyclic scaffold, the integrity of which was considered essential for activity. Therefore this work expands the limited scope of non-natural Sec61 inhibitors to simplified analogues of synthetically-expedient Ipom-F. 

The authors carry out established in vitro and live-cell assays altogether providing consistent results, and further study the interaction of both macrocyclic and acyclic derivatives with a model of Sec61a in its inhibited state following a previous docking protocol. They subsequently rationally design and synthetize a simplified analogue as an effective lead.

The manuscript is clearly written, well structured, and the methodology is rigorously described. I recommend addressing only the following minor aspects, concerning the molecular docking studies:

Major concerns:

1.      The authors present as key finding that Ipom-F analogues can access the Sec61a channel via multiple binding sites, namely the known BS for mycolactone and/or the lateral gate, based on docking results revealing either one or two BS preferences for each compound. These differences are intriguing given that the studied compounds differ by subtle structural features (e.g., in cyclic Ipom-F vs comp. 22, methylene group vs oxygen atom; or for acyclic analogues 1 vs 5, methylene vs N-methyl group). In cases where a single site was identified, in fact no other site was populated in the calculations (1st docking stage), or is this based on a cutoff for the predicted binding affinity, i.e. the other site was sampled though associated with significantly higher energy? Please clarify the criteria as this is not mentioned in the Methods section (p. 17).

2.      The authors note that the predicted binding affinities lack accuracy to be compared with experimental data as a result of methodological limitations. Nevertheless, the number of contact residues within the CL4 region nicely reflected the rank order of activity, but no further rationalization based on structural features of the inhibitors and their conformations / interactions with specific residues is provided. Indeed, the discussion on docking results is much focused on the positioning of the inhibitor in the Sec61a channel but a molecular-detail view could provide insights into the differences observed for the studied compounds. This also stands concerning the postulated role of the disaccharide moiety, or the 11S chiral center.

Minor points:

1.     p. 6, l. 240: “7.82±0.2 kcal/mol” -> “-7.82±0.2 kcal/mol”.

2.     p. 6: “transmembrane (TM) helices” defined in l. 237 but used previously in l. 229.

3.     Scheme 1: relevant reagents / solvents are missing for a few transformations.

4.     Scheme 1: transformations from 25 to 27 should be presented as a single step as intermediate 26 was not purified and characterized.

5.     p. 9, l. 350: “cf. Figures 3 and S4” -> “cf. Figures 5 and S4”?

6.     Image size could be increased (or eventually the orientation changed) in cases where either 2D (figs. 1, 2) or 3D structures (figs. 4 & 5B) are too small to be clearly interpreted. 

7.     figs. 4 & 5B: images from docking calculations have poor resolution.

8.     pp. 10-11, l. 406-407: “Figures 5A-C” -> “figures 6A-C)”

9.     p. 12, l. 437: “(Figures 5D-5E and S3)” -> “(Figures 6D-6E and S3)”

10.  p. 12, l. 464: “An error less than 0.5 Hz are reported” -> “An error less than 0.5 Hz is reported”.

11.  p. 17, l. 710: The authors mention that the final homology model was energy-minimized in 1000 steps with GROMACS but do not indicate which algorithm was employed.

12.  It would help the reader if the compounds were shown in the same order in Figs. 4, S4A, S4B, and S4C. In addition, in Fig. S4, the protein residues (bars) could be colored differently according to the corresponding Sec61a region (TM2, CL4, CL8, TM3, TM7, Plug), since along the text often only the region is mentioned.

13.  Still on fig. S4, the authors mention that, when positioned at the lateral gate, comp. 22 preferentially interacts with Gln294-Val298 (TM7), however a high contact frequency is not shown at that region in the corresponding (1st) panel in Fig. S4B. This may be due to an error in labeling of the panels, or maybe I am misinterpreting the data but please check these images.

14.  p S18: “Fig. S1 (related to Fig. 1)” -> “Fig. S1 (related to Fig. 2)”?

15.  p S20: “Fig. S2 (related to Fig. 1)” -> “Fig. S2 (related to Fig. 2)”?

16.  p S21: “Fig. S2 (related to Figs 2 and 5)” -> “Fig. S2 (related to Figs 3 and 6)”?

Reviewer 2 Report

The authors present well done study of ring-opened analogues of Ipomoeassin F: design, synthesis, in vitro and/or in cellula biological assays, molecular docking. The structures were characterized by 1H and 13C NMR and LRMS(elecrospray). The purity of the compounds was analyzed by HPLC.

The manuscript is well written. The title, abstract, scheme, tables and figures of the manuscript are adequate to the content. The experimental part gives enough details about the experimental procedures.

There is sufficiently supporting documents.

Reviewer 3 Report

Please refer to the attached document.

Reviewer 4 Report

It is a well written manuscript and research was conducted thoroughly. The introduction is relevant and appropriate as sufficient information regarding the topic has been explained. The results present a thorough discussion with relevant figures and tables. Also, this study properly concludes the major findings.

I recommend it for publication after minor revision.

My comments/suggestions are

Author can discuss following points in introduction

Why there is need inhibit Sec61 protein e.g. write possible toxicity associated with its over expression

Methodology

Author can discuss the toxicity evaluation of designed analogues

Do present the visualization in in-silico molecular modeling e.g. which interaction takes part in binding of ligand with protein

Author could discuss the adsorption distribution metabolism excretion toxicity analysis over designed analogues so can predict the toxicity

 There are some corrections needed to be done as per formatting

Author names were not formatted according to journal requirement

Add citation proper citation of article in citation box

Capitalize keywords

Line numbers were not properly assigned

Quality of figure 3 needs to be improved

Can the figure 3 legend be condensed with very precise information only? At the moment this seems to be a discussion rather than a legend

Scheme 1 lacks consistency; some reactions include solvents while others not; same for subscript/superscript

Improve resolution of structures in Fig. 5A

It is good to add abbreviation section. Several words are unexplained

Remove extra space in line no 61

Remove extra word “10 mM” in line number 66

Experimental section needs another look; Grammatical/formatting and incorrect use of symbols/spaces/units exists

Finally, give a through read to avoid the above-mentioned or similar grammatical errors
